# Staphylococcal Scalded Skin Syndrome in Neonates: Case Series and Overview of Outbreaks

**DOI:** 10.3390/antibiotics12010038

**Published:** 2022-12-26

**Authors:** Charlotte M. Nusman, Charlotte Blokhuis, Dasja Pajkrt, Douwe H. Visser

**Affiliations:** 1Department of Paediatrics, Emma Children’s Hospital, Amsterdam University Medical Centers, Academic Medical Center, 1105 AZ Amsterdam, The Netherlands; 2Department of Paediatric Infectious Disease, Emma Children’s Hospital, Amsterdam University Medical Centers, Academic Medical Center, 1105 AZ Amsterdam, The Netherlands; 3Department of Neonatology, Emma Children’s Hospital, Amsterdam University Medical Centers, Academic Medical Center, 1105 AZ Amsterdam, The Netherlands

**Keywords:** *Staphylococcus aureus*, neonate, skin infection, outbreak

## Abstract

Skin and soft tissue infections caused by *Staphylococcus aureus* (*S. aureus*) cover a wide spectrum of diseases in neonates, including staphylococcal scalded skin syndrome (SSSS). We describe a representative case of SSSS in neonatal twins, which despite recurrence showed a mild clinical disease course. This case was part of a small outbreak on a neonatal intensive care unit and therefore exemplifies the existence of neonatal outbreaks with skin and soft tissue infections by *S. aureus.* Diagnosis is generally based on the clinical picture and response to antibiotics, but can be aided by histology and cultures. Sequence-based molecular techniques are available to evaluate typing and virulence of S. *aureus* in outbreak or surveillance settings. The pillars of treatment are antibiotics and supportive care. Methicillin resistance remains a topic of concern, especially in outbreak settings. Our overview of numerous outbreaks of neonatal S. *aureus* skin infections underlines the importance of outbreak management strategies, including screening to identify the source of the outbreak, and limiting exposure through hygienic measures and establishment of physical boundaries.

## 1. Introduction

Staphylococcal scalded skin syndrome is among the most severe in the spectrum of skin and soft tissue infections caused by *Staphylococcus aureus* (*S. aureus*) in neonates. This systemic disease is caused by haematogenous spread of exotoxins leading to desquamation and blistering of the skin, often combined with systemic symptoms such as fever and irritability.

The incidence of SSSS and the virulence of *S. aureus* have both increased over the past decade [1]. Several case series and outbreaks of neonatal skin and soft tissue infections caused by *S. aureus* have been described, but a comprehensive overview of outbreaks is missing. We present a case of recurrent SSSS in prematurely born twins as part of a small outbreak in a level III neonatal intensive care unit (NICU), and provide a comprehensive review concerning the clinical spectrum of this disease and previously reported outbreaks.

## 2. Case Description

After spontaneous premature rupture of membranes, the twins were born prematurely at a gestational age of 30 weeks and 4 days via a caesarean section due to the prolapse of a hand of the first twin. Both newborns were born in good condition with appropriate birth weights for their gestational age and were admitted to a level III NICU with noninvasive respiratory support. Empiric antibiotic treatment (intravenous penicillin and gentamycin) was discontinued on day three as the blood cultures remained negative. On day eight, we suspected sepsis in the secondly born child (twin II) due to increased apneic spells, for which empirical flucloxacillin and amikacin were administered. Due to quick recovery, unremarkable inflammatory markers, and negative blood cultures, antibiotics were discontinued after 48 h.

On day 10, both newborns developed areas of desquamation around the nostrils, mouth and on both cheeks underneath the high flow nasal cannula fixation materials, with erythema and yellow crusts (Figure 1). Within hours after onset of the first skin symptoms, twin I showed increasing superficial desquamation without crusting, localized behind the ears, on the limbs, on monitor electrode sites on the chest, and in the inguinal folds (Figure 1, panel A–C). Mucous membranes were spared. Twin II was less severely affected, with desquamation limited to the face, left inner elbow, and on the back after removal of a lumbar puncture patch (Figure 1, panel D–F). In both infants, generalized erythema was mild and mostly restricted to flexor folds, and the Nikolsky sign was positive.

The conditions deemed most likely in the differential diagnosis at this time included SSSS, impetigo bullosa, or candidiasis, as the mother received topical miconazole for a suspected candida infection of the breast at the same time. Both infants were initiated on intravenous flucloxacillin and fluconazole. Twin I required morphine for pain management. There were no other signs of systemic illness such as apneic spells, dehydration or feeding problems.

Culture of a skin lesion from the face of twin I showed a methicillin-sensitive *Staphylococcus aureus*, positive for exfoliative toxins (ET) type A and B. *Candida* species were not detected. *S. aureus* was also cultured from oropharyngeal and nasal swabs; microbial cultures from the umbilical and perineal area were negative. The positive skin culture could represent a localized skin infection with progression to SSSS, or contamination. Cultures for twin II showed only commensal organisms. Fluconazole was discontinued and both infants were treated with intravenous flucloxacillin for 10 days. After initiation of antimicrobial therapy, existing lesions healed fully over the course of several days and no new lesions formed. The twins were transferred to the special care neonatal unit of a level II hospital on day 15.

On day 23, desquamation without systemic illness recurred in both infants. Due to the extent of skin lesions in twin I, she was admitted to the NICU for three days; twin II did not require readmission. In twin I, blood inflammatory parameters were unremarkable (C-reactive protein: 0.4 mg/L; leukocyte count 13.9 × 10^9^ /L). Skin and throat cultures again yielded *S. aureus* with an identical antibiogram to the previous culture. A paediatric dermatologist was consulted to confirm the diagnosis and recommended histological examination of a snap-frozen biopsy of a blister roof, which showed only horn lamellae of the stratum corneum and acantholytic epithelial cells (acanthocytes) of the granular layer of the epidermis (Figure 2). No inflammatory cells, micro-organisms, or apoptotic cells were present in this specimen. Findings are consistent with the systemic cleaving effects of the ETs of *S. aureus* in SSSS. Both infants recovered fully after intravenous flucloxacillin for 10 days and adequate pain management. Outpatient follow-up visits have been unremarkable up to 6 months after discharge from the level II hospital.

Within a period of 12 days, another two neonates on the same NICU suffered from SSSS. An outbreak management team (OMT) was established, consisting of two employees of the department Infection Control, a medical microbiologist and the NICU floor manager. The OMT set up an action plan within two weeks, which focused on measures to contain the outbreak, including physical boundaries, colonization screening and hygienic measures. Physical boundaries consisted of extra space between the beds of patients, and personal protective equipment for personnel caring for affected patients. During the outbreak, extra training was provided on hygienic measures, mainly on adequate hand hygiene and intensified monitoring of everybody’s compliance. At time of the outbreak, all medical staff and neonates on the ward were screened for S. *aureus* carriership. Based on molecular typing of the bacteria found in carriership screening and neonatal infections (including but not limited to skin infections), three clusters within the period of 12 days where identified, consisting of one cluster of two patients (patients in case description), and two clusters of both one patient and one healthcare professional. The clusters were not related to each other. No other sources of the outbreak, such as environment/materials or family members of the cases were identified.

## 3. Review

### 3.1. Clinical Picture

The incidence of skin and soft tissue infections caused by *S. aureus* in neonates is hard to specify because of its broad clinical and microbial spectrum. SSSS, as a specific condition in this spectrum, has an annual incidence varying between 0.09–0.52 cases per million inhabitants [2,3]. Incidence in infants under, respectively 1 and 2 years of age is reported to be 25.1 and 45.1 cases per million [1,4]. SSSS is caused by *S. aureus*-specific ETs that cleave the protein desmoglein (Dsg)-1 in the superficial epidermis, causing disruption of keratinocyte adhesion and subsequent intra-epidermal splitting at the level of the granulosa layer (Figure 3) [5,6,7]. Approximately 5% of *S. aureus* strains are capable of producing ETs, either ETA, ETB or both [8]. Haematogenic spread of one or both of these toxins leads to the typical skin symptoms of SSSS: generalized erythema, blistering, and desquamation, with a predilection for friction zones and sparing mucous membranes. Desquamation without prior blister formation, such as described in the twins above, has also been reported in newborns and is termed the scarlatiniform variant. A typical finding upon physical examination is that skin easily sloughs with minimal oblique pressure, which is termed the positive Nikolsky sign. The skin symptoms may be preceded by a prodromal phase of irritability and poor feeding. Fever, leukocytosis, or elevated C-reactive protein levels are infrequently reported in newborns [9].

The primary site of infection is not always determined in SSSS. In most reported cases, *S. aureus* is cultured from distant sites, such as the nose, throat, umbilicus or perineum. Blood cultures are usually negative, with a positivity rate as low as 3% in a small retrospective study [9]. Blister contents are typically sterile, but may yield the causative organism if the primary site of infection is the skin (i.e., impetigo bullosa, see below) or by contamination. Infected mother’s milk has occasionally been reported as a suspected source of infection [10]. Both infants in our report received their mother’s milk, but this was not interrupted or analyzed at the time, as they were both already responding well to treatment. In retrospect, analysis could have provided more information about the source of infection and recurrence [11].

The course and severity in neonates is variable, with extent of blistering and desquamation ranging from small areas in the flexor folds to total body surface desquamation [12,13], and signs of systemic infection varying from none to fulminant sepsis with respiratory and circulatory failure.

### 3.2. Differential Diagnosis

The differential diagnosis of blistering and/or desquamating skin diseases in neonates includes impetigo bullosa, toxic epidermal necrolysis, congenital syphilis, pemphigus, and epidermolysis bullosa.

Impetigo bullosa is a localized skin infection with blistering and desquamation caused by the same *S. aureus* ETs as found in SSSS. In impetigo bullosa, culture of open blisters yields the causative organism, and the ETs are produced locally. There are no signs of systemic illness and Nikolsky’s sign is negative. Impetigo bullosa may progress to SSSS by spread of ETs to the bloodstream as the skin barrier is compromised [14]. One study found that impetigo bullosa is more commonly associated with the presence of ETA, whereas SSSS is more frequently associated with ETB, possibly due to lower levels of anti-ETB antibodies in the general population [15].

Congenital syphilis can affect multiple organ systems and the majority of cases present between 3–8 weeks of age. Skin manifestations are typically maculopapular or vesiculobullous with desquamation of the hands and feet [16]. Other signs can vary and be nonspecific, including low birth weight, rhinitis, hepatosplenomegaly, anemia, thrombocytopenia, abnormal liver function tests, and neurological symptoms [17]. Confirming the diagnosis is challenging due to limited sensitivity of most methods; suggestive findings include the presence of neonatal treponemal-specific IgM, higher non-treponemal antibody titres in neonatal as compared to maternal blood, and detection of *Treponema pallidum* by PCR or dark field microscopy [18].

Toxic epidermal necrolysis is a severe skin reaction most often seen in response to a prescribed drug. After a prodromal phase with flu-like symptoms, red-purple macules spread from the trunk to the rest of the body and progress into blisters. Nikolsky’s and Asboe-Hansen (lateral extension of bullae upon pressure) signs are generally positive. A clinical distinction from SSSS is that the mucous membranes are affected in toxic epidermal necrolysis. Histological examination shows full-thickness epidermal necrosis and a subepidermal split, in contrast to the intra-epidermal split in SSSS. It has a high mortality of 25–30%, with early withdrawal of the potentially causative agent being a key prognostic factor.

Pemphigus is caused by autoantibodies against Dsg, which may cause neonatal sequalae by placental transmission of these antibodies [19]. In pemphigus foliaceus, the antibodies are directed against Dsg-1, but presence of intra-epidermal immunoglobins distinguish it from SSSS. Neonatal sequelae are rarely seen, possibly due to low placental transmission of these autoantibodies, or due to co-expression of Dsg-3 in the neonatal epidermis [20]. Pemphigus vulgaris, with antibodies directed against Dsg-3, does occasionally lead to transient neonatal pemphigus, but this has a distinct clinical presentation with deep epidermal skin splitting and involvement of the oral mucosa.

Epidermolysis bullosa describes over 20 subtypes of hereditary mechanobullous diseases, with distinct clinical presentations varying from localized blistering and hyperkeratosis, to debilitating wounds with extensive scarring and contractures. These can be broadly categorized, and distinguished from SSSS, by the level of blister formation: simplex (intra-epidermally at the basal membrane), junctional (at the epidermal-dermal junction), and dystrophic (intradermally) [21].

In summary, SSSS can often be identified by detailed physical examination, including distribution of skin lesions, sparing of mucous membranes, positive Nikolsky’s sign, and absence of other findings such as hepatosplenomegaly. When the diagnosis is unclear, it can be further distinguished from other exfoliative skin conditions by a complete blood count, cultures from the affected skin and other potential colonization sites, and histological examination of the affected skin.

### 3.3. Diagnostic Techniques

SSSS is primarily a clinical diagnosis, but in case of an unclear clinical picture, insufficient response to therapy, or an outbreak of *S. aureus* skin infections, the diagnosis can be confirmed by molecular typing and histological examination. The latter, as exemplified in Figure 2, is based on evaluation of involved skin layers. Molecular typing includes numerous techniques as depicted in Figure 4 and elaborated on below. In daily practice, the determination of methicillin resistance and ETs are clinically most relevant.

All molecular typing techniques require DNA extraction, which consequently can be analyzed for its “bands” or its “sequence”. Due to lack of standardization of analysis protocols and nomenclature for the band-based pulsed-field gel electrophoresis (PGFE), sequence-based typing of *S. aureus* is currently favored.

Amplified fragment length polymorphism (AFLP) is a combination of band-based and sequence-based techniques [22]. A selection of DNA fragments with markers attached is first amplified by PCR, then visualized on agarose gel electrophoresis to detect banding patterns characteristic for *S. aureus.* The selective amplification step in AFLP improves its reproducibility as compared to PFGE [23].

Due to novel and more rapid analysis techniques, whole genome sequencing (WGS) can be used to identify *S. aureus* and more specifically MRSA [24]. In order to prevent inappropriate precautious actions, WGS should only be used as an adjunct to epidemiological investigation [25].

Most sequence-based techniques require an initial polymerase chain reaction (PCR). The most common proteins used for typing within *S. aureus* include ETs, Staphylococcal protein A (spa), Panton Valentine Leukocidin (PVL) and the Staphylococcal chromosome cassette (SSCmec) [26]. PVL is a cytotoxin which attacks leukocytes by creating pores in these membranes, leading to apoptosis. Although PVL is mostly associated with methicillin resistant *S. aureus* (MRSA), some subtypes of methicillin susceptible *S. aureus* can produce PVL as well [27].

MRSA owe their lower affinity for antibiotics to a variant of penicillin-binding-protein encoded by the mecA-gene, which is found on the SSCmec [28]. MRSA isolates are typically recognized by SSCmec-type, or by multilocus sequence typing, which identifies several key loci per organism by means of PCR and combines these to a sequence type [29,30].

Spa typing exists since 1996 and is based on the order and number of repeated sequences in a single locus polymorphic gene region, containing short repeated sequences [31]. Currently, over 19,000 spa types have been identified in 143 different countries.

### 3.4. Management

#### 3.4.1. Treatment

SSSS can be caused by MSSA as well as MRSA. The causative strain of *S. aureus* in our report was susceptible to flucloxacillin as first-line therapy, as well as erythromycin and clindamycin, and resistant to penicillin. As prevalence of methicillin-resistance varies greatly per region, regional data should inform local guidelines concerning empirical therapy [32]. Several paediatric SSSS case series report varying resistance against clindamycin, macrolides, and trimethoprim-sulfamethoxazole, but consistently low resistance rates against vancomycin [33,34,35]. Interestingly, some paediatric SSSS case series report clinical improvement regardless of susceptibility of the causative strain to the initiated antimicrobial therapy, warranting further investigation of the host immune response against the exfoliative toxins in paediatric and neonatal SSSS [32,33].

Based on current perspectives, in areas with low MRSA prevalence (such as in our reported cases from The Netherlands), empiric therapy with a penicillinase-resistant penicillin like flucloxacillin or nafcillin should be sufficient. In case of allergy, a first or second generation cephalosporin such as cefazolin may be used. Vancomycin should be added in regions where MRSA is highly prevalent, or in cases where first-line therapy fails. Clindamycin is sometimes used as adjuvant therapy due to its inhibiting effects on bacterial toxin production. However, recent studies show added clindamycin did not lead to faster recovery as assessed by length of hospital admission [33,35]. Based on the lack of benefit and the high resistance rates, we would recommend against the use of clindamycin as monotherapy or adjuvant therapy in neonates with SSSS.

Initial administration of antibiotics should be intravenous; when patients have improved clinically and there is no bacteremia this can be switched to oral administration. The total treatment duration varies from 5 to 14 days, depending on clinical recovery and microbiological testing, including susceptibility to antibiotics. Topical antibiotic agents such as fusidic acid or mupirocin are useful in more superficial and mild skin infections, and less so in SSSS where the primary site of infection is not necessarily the skin.

Supportive care is crucial in the treatment of skin infections in neonates. Firstly, disrupted skin integrity causes fluid loss, often requiring intravenous fluids in addition to optimal enteral feeding. Secondly, skin care includes gentle handling of intact skin and wound care with non-adherent (fatty) dressings. The third pillar in supportive care consists of adequate pain relief.

In life-threatening disease, most likely to occur in preterm infants or otherwise immunocompromised patients, intravenous immunoglobulins may have beneficial effects based on limited reports [36,37]. Future treatment strategies include the development of anti-virulence therapies, targeting the inhibition of toxin production by *S. aureus* [38].

SSSS generally has a favorable prognosis with full recovery of skin lesions without scarring, although temporary hyperpigmentation is reported in neonates with pigmented skin. Recurrence, such as described above, is occasionally reported in neonates [39,40]. The mortality rate of 3–4% is considerably lower than the reported rates of >50% in adults, which can partially be explained by a higher prevalence of underlying illness in the latter group [6].

#### 3.4.2. Outbreak Containment

Outbreaks of skin and soft tissue infections in neonates caused by *S. aureus* are commonly reported. These outbreaks on neonatal intensive care units and maternity wards are caused by a combination of the relative immunocompromised status of neonates and high risk of transmission of *S. aureus* through asymptomatic carriers [41]. In Table 1 we provide an overview of all outbreak reports (*n* = 23) on *S. aureus* skin and soft tissue infections by neonates between 1961 and 2019. Host and pathogen characteristics from the outbreak reports are summarized below.

The control measures as extracted from these reports are summarized in Figure 5. Currently the only published guideline for management outbreaks of *S. aureus* in a NICU is a consensus statement by the Chicago Department of Public Health in 2006 [42]. This statement focuses on MRSA and provides a list of 27 practical recommendations, rated by the existence of supporting evidence.

**Table 1 antibiotics-12-00038-t001:** Host characteristics in the overview of reported outbreaks (*n* = 23) on neonatal skin infections by *S. aureus*.

Author	Country ^a^	Year	Patients (*n*)	Time Period (Months)	Age Range ^b^ (Days)	Gestational Age ^c^ (Weeks)	SSSS ^d^	Intensive Care ^e^	Recurrence	MSSA/MRSA	Source ^f^
Nusman (this report)	NL	2019	4	0.5	7–10	31.9 ± 5.4	3/4	1/4	2/4	MSSA	N/S
Gopal Rao [43]	UK	2019	12	2	-	31.7 ± 2.7	0/12	1/12	-	MSSA	U
Pimentel de Araujo [44]	IT	2018	12	8	3–17	-	10/12	-	-	MSSA	U
Lamanna [45]	IT	2017	3	2	3–9	-	3/3	-	-	MRSA	P/S
Lee [46]	KOR	2014	6	1	6–14	-	0/6	1/6	-	MRSA	U
Sanchini [47]	IT	2013	9	1	4–27	-	0/6	0/6	-	MRSA	E/N/S
Paranthaman [48]	UK	2012	8	4	6–10	>37	8/8	0/8	-	Both	S
Alsubaie [49]	SAU	2012	13	1	2–16	38.2 ± 1.6	0/13	0/13	0/13	MRSA	U
Neylon [50]	IRE	2010	10	2.5	5–13	38.6 ± 1.1 ^g^	5/10	0/10	-	Both	U
Gould [51]	SCO	2009	7	6	4–48	-	5/7 ^h^	0/10	-	MRSA	U
Kurlenda [52]	POL	2009	6	1	5–34	>37	4/6	0/10	-	MSSA	U
James [53]	USA	2008	11	8	4–23	>37	0/11	0/10	-	MRSA	N/S
El Helali [54]	FRA	2005	13	3	4–18	39.1 (37–41)	13/13	1 ^i^	-	MRSA	N/S
Weist [55]	GER	2000	10	4	4–9	>37	0/10	0/10	-	MSSA	E
Zafar [56]	USA	1995	22	6	3–29	>37	0/22	-	0/22	MRSA	N/S
Dave [57]	UK	1994	30	3	-	-	21/23	-	9/30	MRSA	E/S
Richardson [58]	UK	1990	12	1.5	3–17	-	4/11	-	-	MRSA	S
Dancer [59]	UK	1988	12	2	3–16	-	12/12	-	-	MRSA	P/S
Kaplan [60]	USA	1986	A = 84	32	-	-	2/84	-	-	-	E
			B = 51	11	-	-	0/51	-	-	-	E/N/S
Dowsett [61]	UK	1984	13	2	-	-	9/13	-	-	MRSA	U
Curran [62]	USA	1980	68	4	4–15	-	68/68	-	-	-	U
Faden [63]	USA	1976	4	0.25	4–7	-	1/4	0/4	0/4	MRSA	S
Benson [64]	UK	1962	12	1	2–11	-	4/12	0/12	0/12	-	P/S
Howells [65]	UK	1961	14	3	4–10	-	1/14	1/14	-	-	P/S

^a^ FRA = France, GER = Germany, IT = Italy, KOR = Korea, NL = the Netherlands, POL = Poland, SAU = Saudi Arabia, SCO = Scotland, UK = united kingdom, USA = United States of America, ^b^ age range at presentation of symptoms. ^c^ mean ± SD. ^d^ number of neonates affected with SSSS. If not affected with SSSS, neonates had other skin/soft tissue infections. ^e^ Intensive care = respiratory and/or circulatory insufficiency warranting ventilation or pressor support. ^f^ E = environmental, N = neonates, P = parents or caregivers, S = staff, U = unknown ^g^ only from SSSS cases. ^h^ not classified in article itself, number based on blistering and presence of exfoliative toxin(s). ^i^ at least one patient required intensive care (IC) based on a figure with legend, but it was not stated that the other 12 did not require IC.

#### 3.4.3. Host Characteristics

Although patients <3 months of age were included in our search, the population reported in the outbreaks is often less than two weeks of age. Prematurity is seldom reported in the outbreaks [43], but infants born prematurely are more prone for development of SSSS, as illustrated by numerous case reports [11,36,39,40,66,67,68,69,70,71,72,73,74]. The etiology of this vulnerability consists of an immature immune system and impaired renal clearance of exfoliative toxins. Further, preterm neonates have a significantly lower level of maternal anti-ETA-antibodies compared to term neonates [37]. Newborns may also be partially protected from loss of Dsg-1 by co-expression of Dsg-3 in the superficial epidermis, which is not affected by *S. aureus* ETs, and only present in the deep epidermis in adults. The interplay between these factors may account for some of the reported variability in the clinical course of SSSS in infants. As exemplified by our twin case, severe systemic disease is uncommon. Intensive care requirement was reported in only 4/430 cases, which was unfortunately too little to discover any risk factors Furthermore, a wide spectrum of skin and soft tissue infections is described, including pustules, abscesses and blistering or desquamating disease. Recurrence of SSSS, as we describe, is rare and exclusively reported in preterm [39,40,66] or otherwise immunocompromised patients [75]. The only exception was a 1994 outbreak in the United Kingdom, with a recurrence rate of 30% of unknown cause [57].

#### 3.4.4. Pathogen Characteristics

The large time span of the included outbreaks undoubtedly includes heterogeneity in pathogen characteristics, with evolving *S. aureus* isolates and diagnostic techniques. The last five years, molecular typing in outbreaks of skin and soft tissue infections in neonates is predominated by spa typing. Furthermore, methicillin-susceptibility remains the most frequently reported characteristic of *S. aureus.* Most reported outbreaks on skin and soft tissue infections in neonates concern MRSA, but outbreaks with MSSA only have been described as well (Table 1). Another virulence factor of *S. aureus* is the presence of the PVL gene. Of the outbreaks on skin and soft tissue infections in neonates, MRSA and PVL coexist in three reports [46,47,53]. Of note, PVL-positivity has been described in MSSA isolates as well [43,50].

## 4. Conclusions

Our case report review of outbreaks of SSSS in neonates shows that SSSS is mostly associated with a mild disease course, with recurrence only in a minority of cases. Diagnosis is primarily based on physical examination and response to antibiotics, but can be further supported by cultures and histological examination. Recommended empiric antimicrobial treatment is a penicillinase-resistant penicillin or cephalosporin; vancomycin should be added where MRSA is highly prevalent or if first-line therapy fails. There is insufficient evidence to recommend clindamycin as adjuvant anti-toxin therapy. Host responses against exfoliative toxins may explain clinical improvement despite antimicrobial resistance, and should be further investigated. Supportive therapy includes pain relief and management of fluid losses.

In outbreak or surveillance settings, a wide range of molecular techniques can be used to identify *S. aureus* strains and determine virulence. General measures, including hygienic measures, establishment of physical boundaries such as isolation protocols, and screening of personnel may be used in outbreak management. In many of the reported outbreaks in neonates, the source remains unknown, which hampers specific evidence-based recommendations for outbreak containment on the NICU and warrants further research.

### Literature Search Strategy and Selection Criteria

We searched Medline (Pubmed) and EMBASE with the following keywords and MeSH terms: “infant, newborn [MeSH]”, “neonate”, “staphylococcus *aureus* [MeSH]”, “skin”, limiting to English-only articles and screening references of relevant papers. We selected articles concerning outbreaks of *S. aureus* skin infections in neonates by title/abstract, including ages up to 3 months post-term, and excluding articles concerning colonization and infected mothers.

## Figures and Tables

**Figure 1 antibiotics-12-00038-f001:**
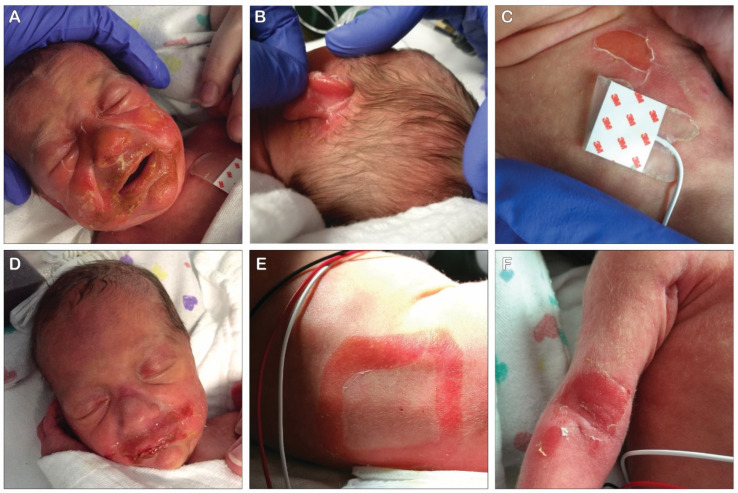
Skin lesions caused by *S. aureus* infection. Skin lesions were more widespread in twin I (panel (**A**–**C**)) as compared to twin II (panel (**D**–**F**)). In twin I, desquamation most severely affected the face (**A**), as well as friction areas such as ears (**B**) and inguinal folds, and skin exposed to adhesives sch as monitor electrodes on the chest (**C**). In twin II, desquamation was limited to sites exposed to adhesives on the face (**D**), back (**E**), and elbow fold (**F**). (Of note: written consent was obtained from the parents to publish these images).

**Figure 2 antibiotics-12-00038-f002:**
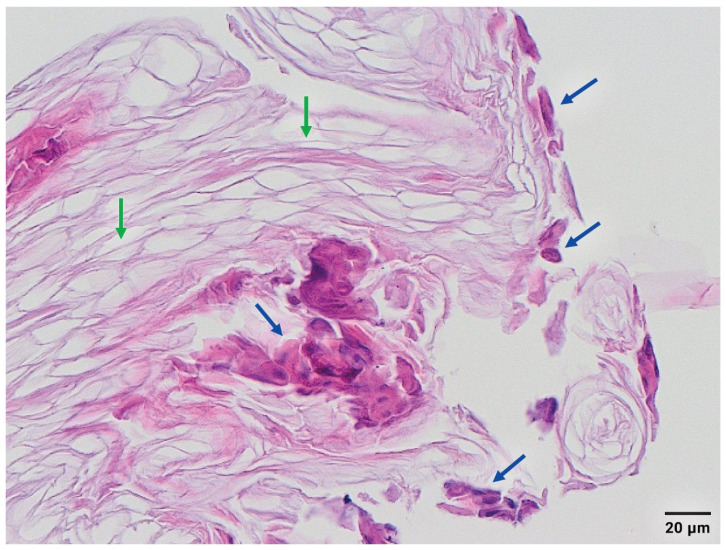
Histology of staphylococcal scalded skin syndrome. Microscopical picture of the snap-frozen blister roof biopsy, showing only horn lamellae of the stratum corneum (green arrow) and some acantholytic epithelial cells (blue arrows) of the granular layer of the epidermis, compatible with SSSS.

**Figure 3 antibiotics-12-00038-f003:**
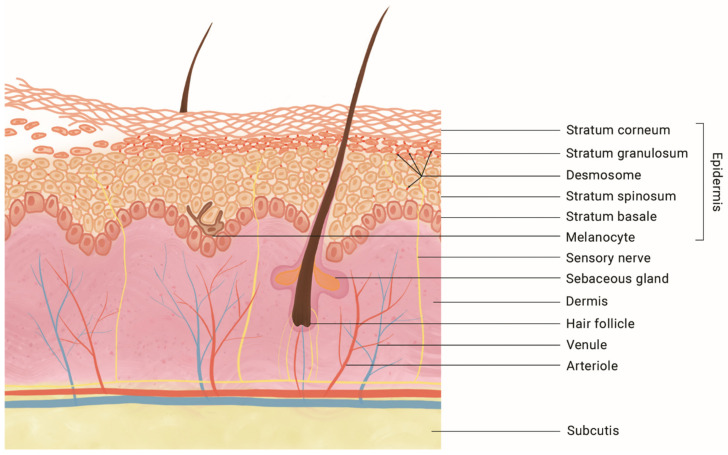
Schematic overview of intra-epidermal skin splitting in staphylococcal scalded skin syndrome. The exfoliative toxins produced by *Staphylococcus aureus* cleave the desmosomal protein Desmoglein-1 (in red, see arrows), causing acantholysis at the level of the granular layer (as pictured on the left). The toxins do not affect Desmoglein-3, which retains its adhesive function in the deeper epidermal layers, as well as in mucous membranes (not pictured) [5,6,7].

**Figure 4 antibiotics-12-00038-f004:**
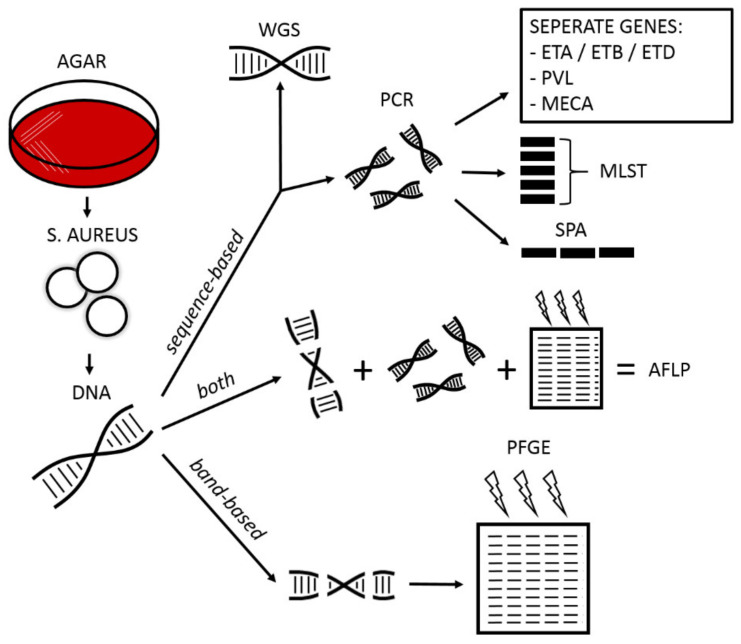
Typing of *S. aureus*. AFLP = amplified fragment length polymorphism; AGAR = agar plate; DNA = deoxyribonucleid acid; ETA/ETB/ETD = exfoliative toxin A/B/D; MECA= gene coding for methicillin resistance; MLST = multilocus sequence typing; PCR = polymerase chain; PFGE = pulsed-field gel electrophoresis; PVL = Panton Valentine Leukocidin; SPA = staphylococcal protein A; WGS = whole genome sequencing.

**Figure 5 antibiotics-12-00038-f005:**
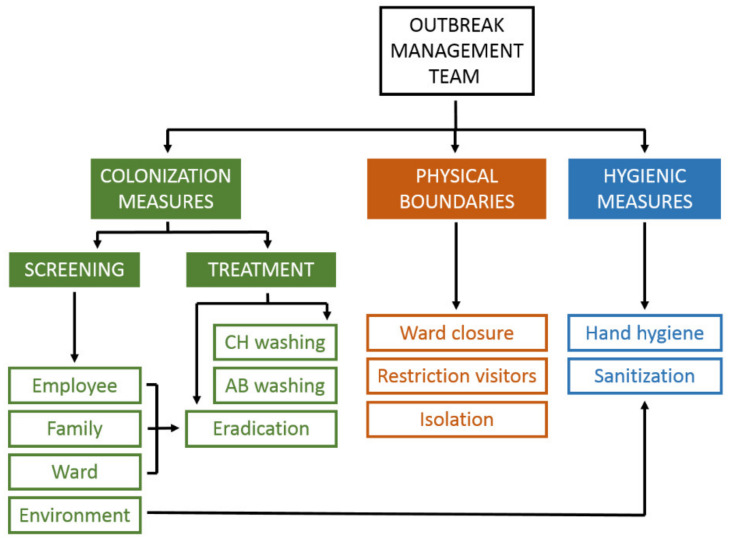
Outbreak containment strategies.Colonization measures and physical boundaries are especially relevant in case of methicillin-resistant *Staphylococcus aureus*. CH = chlorhexidine; AB = antibiotic.

## Data Availability

The data presented in this study are available on request from the corresponding author. The data are not publicly available due to patient-related privacy reasons.

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
