# Peer review of "Staphylococcal Scalded Skin Syndrome in Neonates: Case Series and Overview of Outbreaks"

_antibiotics, 2022, doi:10.3390/antibiotics12010038_

Round 1

Reviewer 1 Report

Abstract Page 1 lines from 20 to 21

Authors: Diagnosis is based on the clinical picture, but can be aided by histology, and sequence-based molecular techniques are available to evaluate type and virulence of S. aureus in outbreak or surveillance settings.

Reviewer: Diagnosis cannot be based on clinical appearance because SSSS shows a similar clinic to many asfoliative dermatitis of the newborn, epidermolysis bullosa, burn lesions, so this statement needs to be corrected

Lines from 23 to 25

Authors: Our overview of numerous outbreaks  of neonatal S. aureus skin infections underline the importance of outbreak management strategies, including colonization and hygienic measures and establishment of physical boundaries.

Reviewer: What does this sentence mean? What does including colonization mean? Colonization what? Cohortization?

Case description

Page 4 lines from 102 to 103

Authors: …….and establishment of physical boundaries.

Reviewer: Please describe prevention strategies, expand.

Review

Figure Page 4 lines from 125 to 128

Authors: Figure 3. Schematic overview of intra-epidermal skin splitting in staphylococcal scalded skin  syndrome. The exfoliative toxins produced by Staphylococcus aureus cleave the desmosomal protein  Desmoglein-1 (in red, see arrows), causing acantholysis at the level of the granular layer (as pictured  on the left). The toxins do not affect Desmoglein-3, which retains its adhesive function in the deeper epidermal layers, as well as in mucous membranes (not pictured).

Reviewer: The figure is not well commented. Putting the cited references in the legend is otherwise incomprehensible

Page 5 Lines from 134 to 137

Authors: ……primary site of infection is the skin (i.e. impetigo bullosa, see below) or by  contamination. Infected mother’s milk has occasionally been reported as a suspected  source of infection Both infants in our report received their mother’s milk, but this  was not tested, as it would not have influenced their treatment at the time.

Reviewer: Please add advice or evidence from breast milk cultures and whether the literature reports the need for milk pasteurization or temporary interruption of breast milk feeding. Why do the authors say that knowing the infectious status of the milk would not have changed the procedures in the face of a cluster of infections? Please explain.

Differential diagnosis

Page 5 lines from 143 to 145

Authors: The differential diagnosis of blistering and/or desquamating skin diseases in neo- nates includes impetigo bullosa, scarlet fever, toxic epidermal necrolysis, pemphigus, and epidermolysis bullosa.

Reviewer: Is it useful to perform VDRL and TPHA between exams? Is it important to evaluate maternal anamnesis in the sense of STDs? Add comment or proposal or literature on this problem

Conclusions page 10 lines from 314 to 330

Authors: Our case report and literature review shows that SSSS is mostly associated with a  mild clinical non-recurring disease course in neonates. Diagnosis is based on clinical findings, but a wide range of molecular techniques can be used to determine S. aureus strains  and virulence in outbreak or surveillance settings. Treatment relies on intravenous anti-  biotics and supportive treatment. Methicillin resistance remains a topic of concern, especially in outbreak settings. Available outbreak management strategies include colonization and hygienic measures and establishment of physical boundaries

Reviewer: The conclusions of a case report with Review must be operational and not generic. How do the authors recommend managing cases of SSSS in neonatology such as diagnosis, therapy and cluster prevention? Please give accurate advice.

In summary

The case is interesting and well documented. There are no details on the laboratory tests performed on the patients that need to be added. Some inaccuracies need to be corrected. Please avoid common sense opinions and replace with concrete suggestions based on literature evidence.

Reviewer 2 Report

This manuscript is fine as a clinical case study, but the link with antibiotics is tenuous in the extreme.  There is no antibiotic resistance data in this area from the cases, but the authors do talk a bit about beta-lactam resistance in Staphylococcus aureus, then fail to link it to the cases.  Moreover, there is much more to antibiotic resistance, in SSSS and other pathologies, than beta-lactam resistance.  It would have been good if the review could have covered resistances pertinent to skin infections.

Reviewer 3 Report

Dear authors,

Thanks for this manuscript in which you outline two recurrent cases of infants (siblings) with SSSS. It is certainly interesting to read, but a first concern is that it is not a systematic literature review, as your title suggests. A thorough pre-defined search strategy, prior protocol registration, title/abstract and full text screening by 2 researchers, bias analysis and systematic reporting of outcome measures are all absent. Therefore, the title should -at the very least- be amended.

Furthermore, you describe the clinical history of 2 neonates with SSSS and briefly mention that they were part of an outbreak. Is there a reason why you do not disclose any details about the outbreak investigation and the measures that have been taken to control the outbreak (other than the general statement in lines 102-103)? That would for sure be interesting. Therefore, I recommend to shift the scope of this manuscript - "case series and report of outbreak investigation in a level III NICU" rather than "case report and systematic literature review."

Other major concerns:

- Line 35-36 "a comprehensive overview is missing." This sentence should be amended. First, this is not a systematic literature review (see comment above) and a quick PubMed search revealed at least >50 case reports/series of SSSS in neonates (in outbreaks or not). There is a systematic review by Mishra et al (PMID 27651848). Is there a reason why you did not cite this paper? Furthermore, in table 1 you describe some host characteristics of the reported outbreaks (without mentioning the source - if known). Could you elaborate what this adds to the current body of knowledge? Eg. what can we learn from this for outbreaks in the future?

- Line 70-72: agree that infections are the main differential dx. However, did you consider any non-infectious etiology (eg. immunologic - maternal antibodies)? 

- Line 72: "Both infants were initiated on IV fluclox": can you outline why vancomycin (or any other anti-MRSA agent) was not added while awaiting culture and susceptibilities?

- Lines 153-162: Scarlet fever is, to the best of my knowledge, unusual in infants, let alone in neonates. Please reflect that in this section. Furthermore, can you outline how you could differentiate this from SSSS - and what therapeutic implication this would have.

- Lines 227-230: Vancomycin + 'MSSA agent' is the preferred regimen in different guidelines- but vancomycin may be omitted if the baseline risk on MRSA is low. Please reference your treatment recommendations in these sentences.

Minor concerns:

- Line 4-5: "Douwe H. Visser and MD PhD" - is there a missing author? 

- Line 29-30: Agree that SSSS is a severe presentation, but there are even worse outcomes (eg. S. aureus septic shock syndrome). Therefore, I recommend to rephrase this sentence.

- Could you clarify in the case report: was a dermatologist involved at any time point? If so, what did this add? If not, why was this not considered?

Round 2

Reviewer 3 Report

Dear authors,

Thanks so much for this revision. I wholeheartedly agree to the shifted scope of the paper. As a first comment, the title "SSSS outbreak in a level III NICU: case series and outbreak investigation" would be more adequate. 

Furthermore, the aspect of outbreak investigation deserves further elaboration; mainly in terms of the text you added in lines 108-117. A thorough description of your investigation will be of benefit to colleagues elsewhere who will face the challenges of SSSS outbreaks in their units. Therefore, please describe:

- Which disciplines participated in the outbreak team?

- In which time frame did they come up with an initial action plan?

- Which hygienic measures needed additional training?

- How were the clusters linked in time?

- Could you come up with other sources that were considered during the analysis?

In line with reviewer 1, congenital syphilis is another infectious differential which incidence is rising in different parts of the world. Thanks for adding lines 170-75. Nevertheless, the part on investigations should be clarified in the text: congen syphilis is usually diagnosed based on the comparison of serology (RPR)  in mother-child dyad In terms of the diagnosis (lines 170-75)- which has many challenges and uncertainties. Other diagnostic methods include PCR on suspected lesions/secretions.

Furthermore, when reading through the text, I noticed some language issues:

- The words 'pediatric' and 'paediatric' are both used throughout the manuscript- please make this uniform. 

- Lines 22, 24 and 113: aureus should be written in Italics.

- Line 24: underline should be underlines.

- Line 39: Please replace "clinical picture" by "case"

- Lines 44-45 should be omitted as this information is shared in the introduction section.

Round 3

Reviewer 3 Report

Dear authors,

Thanks for your continuous efforts to edit this manuscript, the quality has increased significantly. 

Some minor comments:

- Line 106: There should be a space between 'an' and 'outbreak'

- Line 180-81: please add "or dark field microscopy" since that is a valid direct diagnostic method too.

- Figure 4: MECA should be written as mecA

Author Response

The minor comments of the reviewer were all changed accordingly.